# Immunopathologic Role of Fungi in Chronic Rhinosinusitis

**DOI:** 10.3390/ijms24032366

**Published:** 2023-01-25

**Authors:** Seung-Heon Shin, Mi-Kyung Ye, Dong-Won Lee, Sang-Yen Geum

**Affiliations:** Department of Otolaryngology-Head and Neck Surgery, School of Medicine, Catholic University of Daegu, Daegu 42472, Republic of Korea

**Keywords:** fungus, chronic rhinosinusitis, biofilm, extracellular trap, mucosal immunity

## Abstract

Airborne fungi are ubiquitous in the environment and are commonly associated with airway inflammatory diseases. The innate immune defense system eliminates most inhaled fungi. However, some influence the development of chronic rhinosinusitis. Fungal CRS is thought of as not a common disease, and its incidence increases over time. Fungi are present in CRS patients and in healthy sinonasal mucosa. Although the immunological mechanisms have not been entirely explained, CRS patients may exhibit different immune responses than healthy people against airborne fungi. Fungi can induce Th1 and Th2 immune responses. In CRS, Th2-related immune responses against fungi are associated with pattern recognition receptors in nasal epithelial cells, the production of inflammatory cytokines and chemokines from nasal epithelial cells, and interaction with innate type 2 cells, lymphocytes, and inflammatory cells. Fungi also interact with neutrophils and eosinophils and induce neutrophil extracellular traps (NETs) and eosinophil extracellular traps (EETs). NETs and EETs are associated with antifungal properties and aggravation of chronic inflammation in CRS by releasing intracellular granule proteins. Fungal and bacterial biofilms are commonly found in CRS and may support chronic and recalcitrant CRS infection. The fungal–bacterial interaction in the sinonasal mucosa could affect the survival and virulence of fungi and bacteria and host immune responses. The interaction between the mycobiome and microbiome may also influence the host immune response, impacting local inflammation and chronicity. Although the exact immunopathologic role of fungi in the pathogenesis of CRS is not completely understood, they contribute to the development of sinonasal inflammatory responses in CRS.

## 1. Introduction

Airborne fungi are ubiquitous in the environment and continuously inhaled and deposited in the airway mucosa. The number of fungal species exceeds 50,000 with only a few species implicated in human diseases. *Alternaria, Aspergillus, Cladosporium, Penicillium,* and *Candida* are commonly associated with airway mucosal diseases [1,2]. Airborne fungal bioparticles such as spores, mycelia, and hyphal fragments act as allergens that induce type 1 hypersensitivity with the production of specific immunoglobulin E (IgE). Alternatively, they can act as an irritant, causing local or systemic infection, especially in an immunocompromised host. In the air, 50–50,000 spores/m3 are present, and they are continuously inhaled into the airway, where they come into contact with the airway mucosa [3]. Fungi and their components or products contribute to the development of airway inflammatory diseases through innate or adaptive immune responses in airway mucosa.

Fungal rhinosinusitis (FRS) used to be an uncommon disease; however, the number of diagnosed cases has increased over time with the improvement of diagnostic technologies [4]. The rise in FRS may be related to the increased usage of antibiotics, longer life expectancy, global warming, increased air pollution, and increase in the amount of time spent indoors [5,6]. Based on histopathologic findings, FRS can be divided into invasive and noninvasive types. Invasive FRS has histologic characteristics of mucosal infiltration with fungal elements that can be divided into acute fulminant invasive, chronic invasive, and chronic granulomatous FRS. Noninvasive FRS includes saprophytic fungal infection, fungus ball, and allergic fungal rhinosinusitis (AFR) [7]. Fungus ball refers to the accumulation of a dense conglomeration of fungal hyphae in the sinus cavity with or without characteristic hyperdense radiologic findings. In early studies, *Aspergillus* was thought to be the leading cause of fungus ball, similar to bronchopulmonary aspergilloma or allergic bronchopulmonary aspergillosis [8,9]. The definition of AFR is often confused with fungus-related eosinophilic mucin rhinosinusitis [10]. In AFR, Dematiaceous fungi and *Aspergillus* are common etiologic fungi. Typically, AFS is found in immunocompetent atopic patients with chronic rhinosinusitis (CRS) and nasal polyps who develop an allergic response to fungal organisms that have colonized the sinus mucosa with eosinophilic mucus [9]. In some cases, eosinophilic mucin can be found in CRS patients without type I hypersensitivity against fungi [11].

CRS is one of the most frequently reported chronic diseases. In contrast to acute rhinosinusitis, where the bacterial or viral etiology is well established, the etiology of CRS is not completely understood. Noninfectious inflammation with inflammatory cell infiltration of the sinonasal mucosa is a histologic hallmark of CRS [12,13]. Several studies have attempted to elucidate the mechanisms of CRS and numerous hypotheses concerning its pathogenesis have been proposed, such as chronic bacterial infection, superantigens, biofilms, anatomic abnormalities, and immune cell dysfunction [14,15]. In the late 1990s, the Mayo group reported that CRS patients and healthy volunteers had fungal culture-positive nasal secretions. Additionally, they suggested that CRS patients may show abnormal immune responses against airborne fungi [11]. Subsequently, many researchers attempted to detect fungi in nasal secretions in various ways [16,17]. Most fungal CRS are associated with eosinophilic CRS, which develops early olfactory dysfunction and bilateral nasal polyps with opacification of the posterior ethmoid sinus and the olfactory cleft in early CT images in comparison with noneosinophilic CRS [18]. CRS patients have a larger burden of fungi than healthy controls, and sinus surgery can significantly reduce the number of fungal organisms in the sinonasal mucosa [19]. Although CRS patients show abnormal or inappropriate innate and adaptive immune responses against fungi, the role of fungi in the pathogenesis of CRS remains under debate. Because antifungal therapy is not efficacious in controlling CRS, the hypothesis that fungi play a significant etiologic role in CRS development has been rejected [20]. However, many studies have suggested that fungi affect immune responses in the upper airway mucosa. This paper aims to review the recent studies on the immunologic roles and interaction of fungi with the upper airway mucosa and inflammatory cells and its effect on the development of airway mucosal inflammatory diseases or CRS.

## 2. Upper Airway Mucosal Immune Responses against Fungi

The human airway mucosa defends against invasion by environmental pathogens through innate and adaptive immune responses [21]. Innate immunity is nonspecific and includes the activation of the complement system, phagocytosis, and physical and chemical barriers against infectious agents. Most pathogenic organisms are detected and destroyed within hours by innate defense mechanisms. Innate immune responses are followed by adaptive immunity with specialized immunological memory to eliminate or prevent the growth of pathogens and provide long-term clearance of the infection. Airway epithelial cells are essential in the innate immune system as the first mucosal defense against allergens and pathogens. The physical barrier function of the nasal mucosa is determined by the integrity of the apical junction complex, which is composed of tight and adherens junctions [22]. This innate barrier function seems to be altered in chronic airway inflammation. In CRS, the impaired epithelial barrier function leads to increased susceptibility to pathogenic agents and diminished transepithelial migration of inflammatory cells [23]. Fungal components may induce airway epithelial barrier dysfunction with intracellular production of reactive oxygens species and increased epithelial permeability associated with structural and functional alteration in the junctional complexes of the epithelial layer [24,25]. *Alternaria* influences tight junction molecule expression and rarely affects apical junction components in the nasal epithelial cells [25,26]. The imbalance of fungal protease and protease inhibitors within the airway mucosa can cause epithelial barrier dysfunction [27]. Fungal proteases affect the innate immune system of epithelial cells through increased transepithelial access of pathogens and critical immune cells, leading to tissue damage and immune activation [25]. Fungal proteases have immunomodulatory functions by inducing inflammatory cytokines and chemokines in airway epithelial cells, leading to the recruitment, activation, and survival of inflammatory cells [28]. Protease-activated receptors recognize fungal protease in airway epithelial cells. Fungal proteases cleave extracellular ligands, which are freed to initiate the production of inflammatory mediators [29,30]. *Alternaria* induces the production of granulocyte-macrophage colony-stimulating factor (GM-CSF), interleukin (IL)-6, IL-8, and thymic stromal lymphopoietin (TSLP) via the activation of protease-activated receptor (PAR)2, facilitating the development of airway inflammation. Blocking the protease activity of fungi can decrease airway inflammation and hyperresponsiveness [29]. Fungi can also drive airway inflammation and epithelial cytokine production independent of protease-activated receptors [31]. Fungal proteases impair the innate mucosal defense by protein degradation and fungicidal protein activity or by inactivating the complement system [32]. The reduced production of antimicrobial peptides, such as surfactant protein-D, lactoferrin, and histatins, from airway epithelial cells could impair antifungal immunity [33].

Airway epithelial cells actively participate in inflammatory responses by releasing inflammatory mediators that directly affect the airway mucosa and other inflammatory cells. Nasal epithelial cells express pattern recognition receptors (PRRs), which play a crucial role in the proper function of the mucosal innate immune system [34]. PRRs interact with pathogen-associated molecular patterns of fungi, such as glucans and chitin. Toll-like receptors (TLRs) are expressed on various immune cells and non-immune epithelial and endothelial cells and fibroblasts. TLRs play a role in pathogen recognition and in the induction and regulation of innate and adaptive immune responses [35]. TLR2 and TLR4 mRNA expression are significantly increased in nasal epithelial cells of CRS patients [36]. TLR2 interacts with β-glucan and phospholipomannans from fungal conidia and hyphae, and TLR4 can be activated in conjunction with O-linked mannans and Dectin-1 [37,38]. The interaction of fungal allergens with PRRs induces the production of pro-inflammatory cytokines, the respiratory burst, and Th1 immune responses for the antifungal effector function of respiratory epithelial cells. *Aspergillus* and *Alternaria* induce the production of IL-8 and GM-CSF from nasal epithelial cells, which have been implicated in the development of protective immunity against fungi [39].

Mucosal dendritic cells ingest fungal antigens and migrate to the regional lymph nodes, promoting the proliferation and polarization of naïve T-cells. Protective and successful clearance of fungal antigens by Th1 and Th17 immune responses and antifungal responses can be repressed by inhibitory cytokines produced by Th2 cells [40,41]. Th1 responses enhance the functions of phagocytic cells and the promotion of B-cell production of opsonizing antifungal antibodies. Th17 responses activate structural cells, resulting in the production of chemokines that induce the recruitment of phagocytic cells [41,42]. Th2 immunity to fungi is characterized by an inability to clear fungal pathogens with less effective activation of macrophages, and it arises as a consequence of fungal infections [43]. PRRs, such as TLRs and C-type lectin-like receptors, recognize fungal elements and activate epithelial cells to release innate inflammatory cytokines. Fungi-induced epithelial cell derived cytokines, such as IL-25, IL-33, and TSLP, are critical regulators of Th2 immune responses in the nasal mucosa. These cytokines activate group 2 innate lymphoid cells (ILC2) and polarize naïve CD4+ T-cells to Th2 cells at the site of inflammation [44,45]. *Alternaria* induces the production of epithelial cell-derived cytokines through the nuclear factor (NF)-κB, activator protein (AP)-1, and mitogen-activated protein kinase (MAPK) pathways [45]. *Aspergillus fumigatus* also activates the innate immune response and induces the production of IL-33 from nasal polyp epithelial cells through PAR interactions [46,47]. ILC2s play a critical role in fungus-induced immunopathology in airway inflammation. ILC2 induces epithelial repair in the allergic immune response to fungi through the production of the epidermal growth factor [48]. Intranasal instillation of *Alternaria* induces the production of cysteine leukotriene, the activation of ILC2, and Th2 recruitment and proliferation [49]. Several fungal allergens induce type 2 inflammations by enhancing the production of epithelial cell-derived cytokines and potentiating ILC2-associated innate and adaptive immune responses in chronic airway inflammation. The immunopathologic roles of fungi in airway mucosa are summarized in Figure 1.

Recurrent exposure of naïve mice to fungal extracts or in combination with other airborne allergens induces CRS with an increase in airway eosinophils, tissue remodeling, and allergen-specific IgE [50,51,52]. Intranasal instillation of *Aspergillus*-derived proteases or a combined extract of house dust mite, *A. fumigatus*, *Alternaria alternata*, and proteases from *Staphylococcus aureus* successfully induced Th2-type CRS in a murine model with subepithelial infiltration of eosinophils and increased IL-4, IL-5, eotaxin, and other inflammatory mediators in the sinonasal mucosa or nasal lavage fluid [50,51]. The allergic background of the host could influence the immune response against fungi, with allergic predisposed mice showing Th2 and eosinophil dominant immune responses and those without allergic predisposition showing Th2, Th1, and Treg immune responses with eosinophilic and neutrophilic mucosal inflammation [52]. In CRS, fungi induce the innate immune response in airway epithelial cells and ILC2 and the adaptive immune response via B-cells and T-cells, which aggravate type 2 inflammations with severe or intractable chronic airway inflammation. Additional studies are required to determine the effect of host or genetic immunologic backgrounds against fungi on the development of CRS.

## 3. Fungal Biofilms in CRS

Biofilms are microbial-derived sessile communities with microorganisms and extracellular polymeric substances. In human hosts, fungal biofilms can evade host defenses and demonstrate resistance to antimicrobial host immunity and antifungal resistance as a reservoir for recalcitrant infection [53]. Several hundred conidia enter the airway each day, and persistent contact with the airway mucosa can germinate and produce mycelium. In this manner, a biofilm is created via multicellular community formation and interactions with polysaccharides in the extracellular matrix. In CRS patients, the prevalence of biofilm ranges from 44% to 92%. Fungal elements are commonly found within sinonasal mucosal biofilms, especially in eosinophilic mucus CRS [54]. Fungal biofilm colonization of the respiratory mucosa may be an important factor in chronic inflammatory diseases. Inhaled hyphae spread across the mucosal surface with extracellular matrix secretions and can develop fungal biofilms by gluing the hyphae together. In an in vitro study, *A. fumigatus* could develop biofilms on primary nasal epithelial cells [55]. However, in an in vivo study, fungal biofilms could not develop in the sinonasal mucosa with intact innate immune defenses. However, fungal biofilms can develop when innate immune systems are impaired, such as impaired mucociliary clearance or epithelial injury [56]. Impaired mucosal innate defense systems and aggravated mucosal inflammatory responses increase the risk of developing fungal biofilms and fungal inflammatory diseases in the sinonasal mucosa [57]. Preoperative or intraoperative biofilm detection is associated with a high risk of recalcitrance, which requires more aggressive postoperative antimicrobial and anti-inflammatory therapy [58]. The major molecular components that play an important role in the transition from a planktonic form to biofilm form may determine the pathologic characteristics of fungal biofilms.

Microbial biofilm formation from a planktonic form requires interactions between different pathogen and host factors. Bacterial and fungal biofilms commonly co-exist in the sinonasal mucosa of CRS patients [59]. *S. aureus* biofilm is significantly associated with fungal biofilm in CRS. In animal models, inoculation with fungal spores in an obstructed sinus did not develop a fungal biofilm. However, when the fungi were inoculated with *S. aureus*, there was robust biofilm formation in the sinonasal mucosa [56]. Co-inoculation with *S. aureus* and fungi can enhance the formation of fungal biofilms in sheep models, and the degree of mucosal inflammation and mucosal injury was more severe than bacterial or fungal inoculation alone [56]. However, producing fungal biofilms in animal models of sinus mucosa with intact innate immune defenses is difficult. Fungal biofilms developed when *A. fumigatus* was cultured in vitro on nasal or bronchial epithelial cells [57,60]. In CRS, bacterial biofilms in sinonasal mucosa damage the epithelial layer, and then fungal biofilm formation enhanced by aggravated mucosal inflammation and severe recalcitrant disease. Bacterial biofilms may significantly damage mucociliary transport systems that clear inoculated fungal spores. Bacterial and fungal biofilms have a potentially mutualistic relationship. This implies that for the development of sinonasal mucosal inflammation, the interaction between bacteria, especially *S. aureus*, both with colonizing fungi and inhaled fungal allergens, is pathogenic. However, the relationship between the fungus and the bacterium may be synergistic or antagonistic depending on the secretion of bacterial quorum-sensing molecules and host immune factors [61]. The initial interaction between the fungus and bacteria is synergistic. However, during the formation of biofilms, it may become competitive or antagonistic depending on the species of microbe and host immune factors. Biofilms may affect the chronicity and recalcitrant nature of CRS, but further studies are needed to determine the pathophysiologic role of biofilms on the development of CRS.

## 4. Neutrophil and Eosinophil Extracellular Traps against Fungi

Neutrophils and eosinophils act as effector immune cells that mediate antifungal immunity. Extracellular traps are a type of rapid cell death characterized by the release of intact cytoplasmic organelles via nuclear and plasma membrane breakdown [62]. Extracellular traps remove pathogens by capturing and immobilizing them by releasing a mixture of DNA, histones, and granule proteins. Immunologic characteristics of NETs and EETs were summarized in Table 1. Neutrophil extracellular traps (NETs) and eosinophil extracellular traps (EETs) are commonly found in CRS [63,64]. Tissue eosinophilia and EETs or tissue neutrophilia and NETs may affect the prognosis and refractory nature of CRS. Neutrophils are abundant in humans and quickly respond to kill harmful agents by trapping and releasing antimicrobial molecules [65]. NETs provide protection against bacterial infections; for instance, lipopolysaccharide and flagella trigger NET formation with bacteriostatic or bactericidal effects. NETs play a significant role in clearing infections, such as with mycobacteria, fungi, viruses, and parasites [65,66,67,68]. NETs are characterized by morphological changes in nuclear shape, chromatin condensation, and leakage of the nuclear envelope with phorbol-12-myristate-13-acetate-induced cell death. However, NETs can also develop independent of cell death by granulocyte/macrophage colony-stimulating factor, lipopolysaccharide or complement factor 5a, which induce the release of mitochondria-derived DNA with an intact nuclear membrane and neutrophil viability [69]. NETs significantly decrease the extracellular microbial burden to inhibit infection and increase host survival. Bacteria, fungi, viruses, and immune complexes induce NETs through nicotinamide adenine dinucleotide phosphate (NADPH)-oxidase and the reactive oxygen species (ROS)-dependent response of neutrophils with activation of intracellular granular proteases [66]. Neutrophils and their NETs are key players in innate defense against fungi. Neutrophils and other immune cells can remove small fungal structures, such as conidia and yeast through phagocytosis. Fungal hyphal filaments are too large to be removed by phagocytosis, but these may be the main targets of NETs [70]. The size of fungal components and their morphology also influence NETs through various signaling pathways. β-glucan particles or molecules trigger NETs through the Syk kinase-dependent pathway [71]. *A. fumigatus* induce NET formation within 3–4 h after exposure to neutrophils without prior stimulation. Fungal hydrophobin RodA on the surface of conidia with β-glucan inhibit NET production, but the fungal pigment DHN-melanin was not involved in the evasion of neutrophil killing [70]. During the formation of fungal biofilms, the expression of RodA increased compared with the planktonic condition. Fungal biofilms show inhibition of NETs with immune evasion [72]. The number of NETs-forming neutrophils was significantly higher in acute exacerbated CRS patients, meaning that microorganisms including bacteria and viruses trigger NETs to kill microbes or prevent microbial dissemination [63]. However, NETs show antifungal properties and can be pathogenic by being cytotoxic to airway epithelial cells and structural cells, aggravating chronic inflammatory diseases. However, until now, drawing conclusions has been difficult regarding the interactions between fungi and NETs in sinonasal mucosa as playing an important role in development of CRS.

Eosinophils protect the host against nonpagocytable pathogenic fungi by releasing cytotoxic granules [73]. Eosinophil cationic proteins, major basic protein, eosinophil-derived neurotoxin, and eosinophil peroxidase are four major intracellular crystalloid-bearing granule proteins. Lilly et al. demonstrated that eosinophils are essential for fungal clearance and fugal growth suppression through the production of pro-inflammatory cytokines and chemokines [74]. Eosinophils and their degranulation products are abundant in the eosinophilic mucin of CRS patients [75]. EETs can occur either independent of cell death based on the activation of NADPH oxidase and β2-integrin or dependent on cytolytic cell death with extrusion of histone-enriched nuclear DNA associated with clustering of intact granules [76,77]. Direct contact with fungal components, such as fungal cell wall protease, chitin, and β-glucans, initiates eosinophil activation, resulting in antifungal responses [78]. Eosinophils also have fungicidal activity against *A. fumigatus* without direct cell contact [74]. Fungal conidia and eosinophils provide an adhesive surface for microorganism entrapment and EET formation. The eosinophilic mucin of CRS patients contains the extracellular deposition of cytotoxic granule proteins that destroy fungal elements [79]. The increased viscosity of nasal secretions in CRS is not only associated with the overproduction of mucin glycoproteins from airway epithelial cells but also EETs, which contain debris of inflammatory cells and large polymers including DNA [80]. Abundant filamentous DNA fragments in eosinophilic mucin form a sticky scaffold for clustered eosinophils and free eosinophilic granules. EETs provide an adhesive surface for fungal entrapment. Eosinophils form EETs in response to *A. fumigatus* conidia through a cytolytic process with intact granules [81]. Phorbol myristate acetate (PMA) induces EET in an oxidative burst and in a NADPH oxidase-dependent manner. Compared with NETs, which are associated with NADPH oxidase or mitochondrial ROS, EET development by fungi occurs independently of ROS production in leukocytes. These differences between eosinophils and neutrophils in the *A. fumigatus*-induced release of extracellular DNA traps may be related to the difference in how they recognize or respond to fungi. The interaction of an eosinophil β2-integrin molecule, CD11b, and fungal cell wall β-glucan play an important role in the release of EETs [81]. *A. fumigatus* induces EET formation by a cytolytic process via a Mac-1 and Syk tyrosine kinase pathway-dependent mechanism [81]. However, the role of PPRs and the interaction with other structural cells in EET release remain to be studied. EETs can eradicate fungi, although how they interact with the immune system under normal physiological conditions has yet to be determined, with either a beneficial or harmful effect on the development of CRS. EETs are an innate defense system against extracellular pathogens, but they induce barrier dysfunction and contribute to the stability and high viscosity of mucus, which in turn impairs pathogen clearance with inflammatory cell infiltration, biofilm growth, and the potential for chronic diseases [82].

## 5. Mycobiome

Inhaled microbial colonization may play a role in the initiation and maintenance of the inflammatory process of CRS. In the field of mucosal immunity, various studies have elucidated the role of the microbiome for the initiation, adaptation, and maintenance of the mucosal immune responses. Dysbiosis of the sinonasal mucosa microbiota may contribute to the development and exacerbation of chronic inflammatory diseases [83,84]. However, most microbiome or microbiological studies are focused on bacteria, and the characteristics or composition of fungal species in human body are not commonly studied. Like the microbiome, the mycobiome may also play an important immunologic role in host innate and adaptive immunity. Mycobiota are the fungal component of a given microbial community, whereas the mycobiome is their corresponding genomes [85]. Fungi are ubiquitous and normal commensals of sinonasal mucosa in healthy and CRS subjects, but the interactions with the immune system under normal or pathologic conditions has yet to be determined. Fungal culture has been difficult using traditional culture methods. However, the development of next-generation sequencing (NGS) methods and state-of-the art molecular biologic techniques are enabling the detection of fungi by culture-free, fast, and sensitive single-molecule sequencing [86]. Based on traditional methods, *Aspergillus*, *Alternaria*, *Cladosporium*, and *Penicillium* are commonly isolated from sinonasal mucosa [11,87]. With NGS, *Aspergillus* sp., *Schizophyllum* sp., *Curvularia* sp., and *Malassezia* sp. are most frequently detected fungi in the nose [88]. When using nanopore sequencing, *Malassezia* sp., *M restricta* and *M. sympodialis*, are predominant in the sinonasal mucosa of healthy and CRS subjects, followed by *Aspergillus* sp. and *Candida albicans* [89]. Because *Malassezia* is lipid-dependent and is typically found in sebum-rich areas of the body, they require special culture media to grow. The presence of *Malassezia* in the sinonasal mucosa may come from the nasal vestibule. An opportunistic infection by *Malassezia* can damage nasal epithelial cells and induce Th2 immune responses in the sinonasal mucosa [90]. However, the role of *Malassezia* in the development of the CRS is not completely understood. The immunopathologic role of the development of CRS based on this organism requires further study. *A. fumigatus* is often involved in the development of allergic fungal rhinosinusitis and fungus ball. Fungal components are commonly found in the eosinophilic mucin of CRS patients [79]. Alterations in the mycobiome are associated with the development of airway diseases [91]. Fungal dysbiosis is related to the altered fungal composition or changes in total fungal content or abundance. Although fungal diversity should be higher in more severe disease, fungal dysbiosis may be implicated in the development or exacerbation of airway inflammatory diseases through interactions with hose immune responses.

Fungal and bacterial interactions and their relationship with hosts has been studied [92,93]. Bacterial and fungi co-exist in the human body, and microbial interactions influence human health and disease. Bacteria can directly affect fungal morphology, survival, growth, virulence, and attachment [92,94]. Fungal–bacterial interactions could affect the survival and virulence of fungi, bacteria, and both their own and host immune responses. In an in vitro study, *S. aureus* enhanced the tissue invasion of fungi, and *Pseudomonas aeruginosa* showed an antifungal effect [93,95]. Pyocyanin and phenazine, metabolites from *P. aeruginosa*, inhibited the growth of *A. fumigatus*, and *A. fumigatus* suppressed the growth of *P. aeruginosa* [96]. Co-infection of *Malassezia sympodialis* and *S. aureus* demonstrated synergistic interactions in growth and virulence with aggravated type2 and type17 inflammation in the sinonasal mucosa [97]. Interactions between fungi and the host can occur directly or through metabolites produced during colonization and infection. The interactions between fungi and bacteria show species-specific immune responses in the sinonasal mucosa [97]. The downregulation of bacterial loading by antibiotics could influence fungal composition or growth, which may aggravate or suppress mucosal inflammation However, dysbiosis of the mycobiome is primary or secondary to an immune response as an imbalance of the bacterial microbiome. The interaction between the mycobiome and the microbiome may show specific immune responses and influence the host immune response, which may affect local inflammation and disease progression. Therefore, understanding microbial interactions and host-microbiota interactions will be crucial in establishing the function of microbial communities in CRS and implementing new therapeutic strategies.

## 6. Summary and Conclusions

Fungi are ubiquitous in the environment and co-exist with hosts as saprophytes or commensals. Nonetheless, some airborne fungi cause airway inflammatory diseases, and some are associated with the pathogenesis of CRS. Although fungi play a pathogenic role in certain types of CRS, the immunopathologic role of fungi in pathogenesis of CRS is not completely understood. Airborne fungi interact with the sinonasal mucosa by innate and adaptive immune responses. The Th1 immune response commonly associated with antifungal immunity and the Th2 immune response is related to pathologic immunity. However, the host genetic background, airway mucosal fungal loading, fungal species inhaled from the air, susceptibility to fungal allergens, and environmental factors influence the type of mucosal immune responses. Although Th2 inflammation is commonly associated with the pathologic role of fungi in CRS development, Th1 inflammation is also found in the sinonasal mucosa [43,45]. Bacterial–fungal interactions, such as the development of biofilms and the composition of mycobiomes, may also play an important role in determining the inflammatory immune responses in the sinonasal mucosa. Furthermore, their interactions with the host need to be elucidated. In recent years, knowledge has been accumulated through many studies on the immunologic characteristics of CRS, and various immunological mechanisms and hypotheses have been proposed for the role of fungi in CRS development. Although fungi are present not only in CRS or chronic inflammatory airway disease mucosa but also in normal healthy mucosa, they interact with innate and adaptive immune responses to remove pathologic fungi from the sinonasal mucosa or contribute to the development of airway inflammatory diseases. Fungi contribute to the development of CRS, and additional studies are required to reveal a more detailed understanding of the immunologic interactions between fungi, bacteria, and the host sinonasal mucosa.

## Figures and Tables

**Figure 1 ijms-24-02366-f001:**
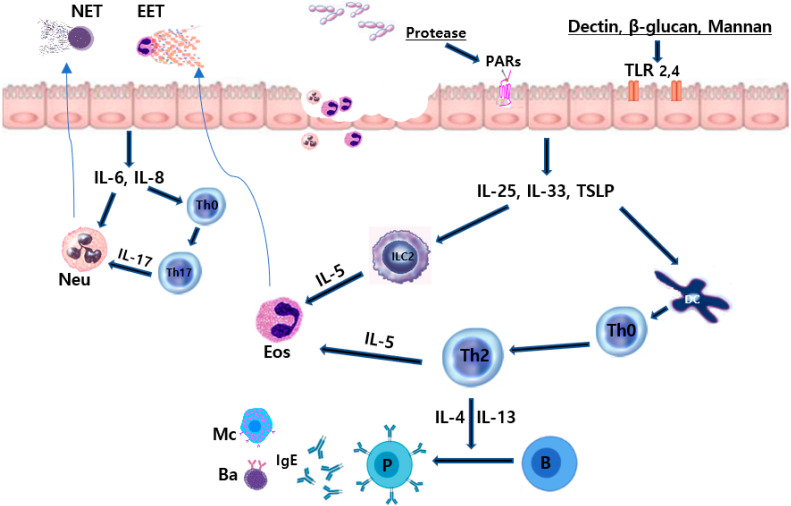
Immunopathologic effects of fungi on the sinonasal mucosa. Fungal proteases and components interact with epithelial cell receptors and release innate inflammatory cytokines. Fungi can induce eosinophilic and neutrophilic inflammation in sinonasal mucosa. IL, interleukin; TSLP, thymic stromal lymphopoietin; IgE, immunoglobulin E; Th, T helper cell; B, B cells; P, plasma cells, Ba, basophils; Mc, mast cells; TLR, toll-like receptor; PAR, protease-activated receptor; Eos, eosinophils; Neu, neutrophils; EET, eosinophil extracellular trap; NET, neutrophil extracellular trap (Authors own work).

**Table 1 ijms-24-02366-t001:** Immunopatholoic characteristics of neutrophil extracellular trap (NET) and eosinophil extracellular trap (EET).

	NET	EET
Stimulants Receptors ET constituents Diseases	Virus, bacteria, fungi, LPS, PMA, platelets, cytokines, Type I interferons, autoantibodies FcγRIIa, TLR-2, TLR-4, C3, C5a, Dectin-1 Nuclear DNA, Mitochondrial DNA Histones, ROS, granule proteinsSystemic lupus erythematous, Rheumatoid arthritis, Gout, Sepsis, Tumor metastasis, Thrombosis, Artherosclerosis, Diabetes	E. coli, LPS, fungi, IgG, IgA Cytokines PMA, Calcium ionophore CD111, TSLPR Dectin-1, β2-integrin Nuclear DNA, Mitochondrial DNA Histones, ROS, granule proteins Atopic dermatitis, Urticaria Chronic rhinosinusitis, Otitis media Chronic obstructive pulmonary disease

LPS, lipopolysaccharide; PMA, phorbol myristate acetate; C, compliment component; ROS, reactive oxygen species; TLR, toll-like receptor, FcγRIIa, IgG Fc receptor IIa; TSLPR, Thymic stromal lymphopoietin receptor.

## Data Availability

Not applicable.

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
