# Peer review of "Immunopathologic Role of Fungi in Chronic Rhinosinusitis"

_ijms, 2023, doi:10.3390/ijms24032366_

Round 1

Reviewer 1 Report

This manuscript aims to review the recent studies on the immunologic roles and interaction of fungi with the upper airway mucosa and inflammatory cells and its effect on the development of airway mucosal inflammatory diseases or CRS.

Some questions should be clarified:

  1. Line 34: type I hypersensitivity
  2. Line 42-43: “The rise in FRS may be related to the increased usage of antibiotics, longer life expectancy, global warming, increased air pollution, and increased in the amount of time spent in- doors “  What evidence does this base on?
  3. Lack of citation in some paragraphs, for example: Line 80-90, Line 117-120, 283-285, 339-349…..
  4. Some citations are old.

Author Response

I thank the editors and referees of the ‘International Journal of Molecular Sciences’ for taking their time to review my article.

I have made some corrections and changed some parts in the manuscript after going over the referee’s comments.

  1. Line 34: type I hypersensitivity

Answer) Authors mistake was corrected as indicated.

  1. Line 42-43: “The rise in FRS may be related to the increased usage of antibiotics, longer life expectancy, global warming, increased air pollution, and increased in the amount of time spent in- doors “ What evidence does this base on?

Answer) We suggested various hypotheses for FRS increase, and reference 6 was added.

  1. Lack of citation in some paragraphs, for example: Line 80-90, Line 117-120, 283-285, 339-349…..

Answer) Line 80-90, Reference 21 & 22 were added.
        Line 117-120, Reference 34 was added.
        Line 283-285, Reference 73 was added.
        Line 339-349, Reference 91 was added.

  1. Some citations are old.

Answer) We tried to cite the latest references, however for the information that required historical backgrounds, old references were cited, such as Ref 7, 8, 10.

Thank you.

Reviewer 2 Report

Hello

A very interesting review. 

Please tell me if figure 1 was drawn by Authors or is it copied from someplace? If drawn by research team, please add to its description (Authors own work);

Did the Authors also investigate papers with FRS 0 fungal-related disease in patients with immune deficiency? Are there any differences between fungal spread in generally healthy persons vs those with immunodeficiency syndromes?

In my opinion 1-2 sentences should be add, it there is any similarities or differences in clinical and raidological symptoms of viral, baterial and fungal sinusitis? - Its important to highlight it in the text

Thank you again for an interesting paper

Author Response

I thank the editors and referees of the ‘International Journal of Molecular Sciences’ for taking their time to review my article.

I have made some corrections and changed some parts in the manuscript after going over the referee’s comments.

Please tell me if figure 1 was drawn by Authors or is it copied from someplace? If drawn by research team, please add to its description (Authors own work);

 Answer) Fig 1 was drawn by our team.
        And ‘Authors own work’ was added at the last part of the Fig 1 legend.

Did the Authors also investigate papers with FRS 0 fungal-related disease in patients with immune deficiency? Are there any differences between fungal spread in generally healthy persons vs those with immunodeficiency syndromes?

Answer) That seems like an interesting question. Most CRS patients have normal immunity. Immune deficiency is commonly associated with invasive type fungal sinusitis.
Personally, I don't have much experience with CRS patients with immune deficiency.

In my opinion 1-2 sentences should be add, it there is any similarities or differences in clinical and raidological symptoms of viral, baterial and fungal sinusitis? - Its important to highlight it in the text

Answer) Clinical characteristics of fungal CRS was described in Line 65-68 as ‘Most of fungal CRS are associated with eosinophilic CRS, which develop early olfactory dysfunction, bilateral nasal polyps with opacification of posterior ethmoid sinus and the olfactory cleft in early CT images in comparison with noneosinophilic CRS [18].’